# Enhancement of Oxidative Stability of Deep-Fried Sunflower Oil by Addition of Essential Oil of *Amomum villosum* Lour.

**DOI:** 10.3390/antiox12071429

**Published:** 2023-07-15

**Authors:** Yunlong Zhao, Haohao Wu, Mengrui Qu, Yuchen Liu, Dongying Wang, Haoduo Yang, Yingying Wang, Xuede Wang, Francesca Blasi

**Affiliations:** 1College of Food Science and Engineering, Henan University of Technology, Zhengzhou 450001, China; 2021920029@stu.haut.edu.cn (Y.Z.); 2021930544@stu.haut.edu.cn (H.W.); 2022920119@stu.haut.edu.cn (M.Q.); 2022920118@stu.haut.edu.cn (Y.L.); 2020920104@stu.haut.edu.cn (H.Y.); 2020920090@stu.huat.edu.cn (Y.W.); 2Department of Pharmaceutical Sciences, University of Perugia, Via San Costanzo, 06126 Perugia, Italy

**Keywords:** oxidative stability, natural antioxidant, flavoring agent, deep-frying, *Amomum villosum* Lour.

## Abstract

In this study, the essential oil of the fruits of *Amomum villosum* Lour. (AVEO) was extracted through steam distillation and the components of the AVEO were analyzed using Gas Chromatography-Mass Spectrometry (GC-MS). Additionally, the antioxidant capacity in vitro of the AVEO was gauged using radical scavenging activity (DPPH, ABTS, superoxide anion) and ferric ion reducing antioxidant power (FRAP) assays; the antioxidant effect of a certain concentration of AVEO is even comparable to 0.08 mg/mL of butylated hydroxytoluene (BHT). Moreover, AVEO was applied to sunflower oil in a 30 h successive deep-frying experiment. Throughout the frying procedure, the sunflower oil-added antioxidant showed different degrees of benign changes in the physical and chemical parameters compared to the blank group, with 1 g/kg of AVEO being more consistent with 0.01 g/kg of *tert*-butyl hydroquinone (TBHQ), while 1.5 g/kg of essential oil revealed a stronger antioxidative capability. Meanwhile, the organoleptic characteristics of Chinese *Maye*, including its appearance, taste, flavor, and overall acceptability, were ameliorated when AVEO was added at 1.5 g/kg. Consequently, AVEO can be applied to substitute synthetic antioxidants as a natural antioxidant and flavoring agent during the deep-frying course of food.

## 1. Introduction

As one traditional food processing method, the deep-frying procedure at 150–200 °C is frequently employed in the production of fried products in houses and some factories [1]. As reported, the deep-frying process can provide foodstuff with a unique taste, golden color, crispy texture, and tempting flavor, making fried products quite popular with consumers all over the world [2]. However, during the deep-frying process, the oxygen gas, high temperature, and release of water from the dough may bring about a sequence of complex chemical reactions in vegetable oils, for instance, lipid oxidation, hydrolysis, polymerization, and isomerization [3]. A considerable number of investigations have demonstrated that these aforementioned chemical reactions can bring about the production of various hazardous substances, including trans fatty acids, acrylamide, polar compounds, and heterocyclic compounds [4]. These hazardous substances often cause the deterioration of vegetable oils and fried products, as well as a variety of serious health problems for humans. In order to effectively reduce the deterioration of edible oils and improve their oxidative stability, abundant research has been carried out. Among them, the direct addition of synthetic antioxidants before the frying process was suggested as an available way to prolong the quality guarantee period of vegetable oils and decrease the production of undesirable components [5]. At present, the main synthetic antioxidants utilized in the food industry to lengthen the quality guarantee period of vegetable oils are butylated hydroxyanisole (BHA), butylated hydroxytoluene (BHT), *tert*-butyl hydroquinone (TBHQ), and propyl gallate (PG) [6], but all of these synthetic antioxidants possess poor thermal stability and low antioxidant efficiency, especially when the temperature is high. More importantly, they indeed have certain toxic side effects on the human body [7]. Unsurprisingly, an increasing number of countries have severely restricted the utilization of synthetic antioxidants. In the meantime, the idea of replacing synthetic antioxidants with natural antioxidants (plant extracts) to retard the oxidative rancidity of vegetable oils has attracted a lot of attention from researchers in this field [8]. Therefore, the search for a safe and stable natural antioxidant from green plants is essential and of great research interest.

The fruits of *Amomum villosum* Lour., also called Fructus Amomi, are one of the most frequently used folk medicines in Chinese medicine. In the 2020 version of Chinese Pharmacopoeia, Fructus Amomi is recognized as the dried and ripe fruit of three plants: *A. villosum* Lour., *A. villosum* Lour. var. *xanthioides* T. L. Wu et Senjen, and *A. longiligulare* T. L. Wu. Among them, *A. villosum* Lour. is supposed to be the dominating source for Chinese medicine, and it has been employed as a “medical treasure” for more than 1000 years [9]. As reported, *A. villosum* Lour. is a kind of herbaceous plant that can grow up to 3 m high in the shade of mountainous areas, which are primarily located in Fujian Province, Guangdong Province, and Yunnan Province in China [10]. Meanwhile, over 100 phytochemicals (saponins, flavonoids, organic acids, volatile oils, etc.) have been extracted from *A. villosum* Lour., which exhibit antioxidant, antimicrobial, antitumor, and other biological effects [11]. Many of the crude extracts from the stems, leaves, and roots of *A. villosum* Lour. possess antioxidant and anti-inflammatory effects [12,13]. However, the main aromatic and medicinal component extracted from *A. villosum* Lour. is essential oil, and the content and activity of the essential oil are considered to be the extremely vital indicator of the quality of Fructus Amomi [14]. Furthermore, the basic ingredients of the essential oil are alkenes, esters, and alcohols; numerous studies have demonstrated that the essential oil has antioxidant, antimicrobial, anti-inflammatory, and analgesic activities at present [15,16,17]. One hypothesis is that AVEO might be employed as a safe and effective natural antioxidant for the deep-frying process of vegetable oil.

Based on the issues mentioned above, this experiment was undertaken with the purpose of investigating the influence of AVEO on the oxidative stability of sunflower oil and the organoleptic attributes of Chinese *Maye*, which is a crispy and delicious fried product, during the deep-frying procedure. Moreover, the chemical components of AVEO were analyzed through GC-MS and the antioxidant capability in vitro of AVEO was determined.

## 2. Materials and Methods

### 2.1. Materials and Chemicals

The ripe and dried fruits of *A. villosum* Lour., originated from Yunnan Province, China, were bought on the online shopping platform Taobao. The fruits were identified by Professor Dongying Wang from the College of Food Science and Engineering, Henan University of Technology, Zhengzhou, China. Cold-pressed sunflower oils without antioxidant manufactured by COFCO Corporation (Beijing, China) were purchased from the network marketplace Jingdong. In addition, the pieces of fresh Chinese *Maye* were bought from Yonghui Supermarket in Zhengzhou, China. TBHQ, BHT, and all the standard chemicals were purchased from Merck (Darmstadt, Germany). Moreover, anhydrous sodium sulfate, hexane, methanol, and other chemicals were of analytical or chromatographic grade, provided by Senbo Biotechnology Co., Ltd. (Zhengzhou, China).

### 2.2. Extraction of AVEO

The extraction of AVEO through steam distillation was carried out based on previously reported methods with some adjustments [18]. First, the dried and mature fruits were crushed using a swinging traditional Chinese medicine pulverizer (XL-16B, Guangzhou, China). For the extraction process, 50 g of sample powder and 1000 mL of distilled water were accurately measured and added in turn to a round bottom flask with a volume of 2000 mL and mixed thoroughly. The flask was placed in a temperature-controlled heating unit; subsequently, the steam distillation apparatus and condenser tube were connected. Then, the heating unit was turned on and adjusted to the suitable temperature (180 ± 5 °C) to keep the flask in a slightly boiling state. When the distillation process was completed, the oily liquid from the upper layer of the essential oil extraction device was collected and dried with anhydrous sodium sulfate to obtain the AVEO. The EO yield was counted in accordance with the mass of the obtained essential oil divided by the mass of the dried plant material (%, *w*/*w*). Finally, the AVEO was rapidly gathered into a brown centrifuge tube and stored in a refrigerator at 2 °C for further research and analysis.

### 2.3. GC-MS Analysis of AVEO

Referring to the previous method with minor modifications, the chemical composition analysis of AVEO was fulfilled through GC-MS [19]. Briefly, an Agilent Technologies 7890B GC System (Palo Alto, CA, USA) was connected to Agilent Technologies 5977B MSD (Palo Alto, CA, USA), installing a HP-5MS capillary column (30 m × 0.25 mm I.D., 0.25 µm F.T., Agilent, USA). The energy of ionization was 70 electron volts. High purity (99.99%) Helium (He) was utilized as a carrier gas at a flow rate of 1.0 mL/min. The injector, with a temperature of 250 °C, and the detector, with a temperature of 290 °C, were operated. The column temperature program was adjusted as follows: the initial temperature was 50 °C, increased from 50 °C to 70 °C by 10 °C/min and held for 2 min, then ramped at the rate of 1 °C/min to 125 °C and held for 2 min, and then increased by 4 °C/min until 210 °C was reached; after a 2 min hold time, the final rise of temperature was increased from 210 °C to 280 °C by 20 °C/min and held there for 5 min. The injection volume of the sample was 1 µL and the split ratio was 1:20. The MS scan ranges were set to between 30 and 500 *m*/*z*. Furthermore, the transmission line temperature was 280 °C and the ionization temperature was 230 °C. A solvent delay of 3 min was selected. Comparing the data obtained from the GC-MS analysis with the National Institute of Standards and Technology (NIST) Chemistry Webbook Database and published literature, the volatile chemical components of the AVEO were positively identified. Then, the peaks of compounds with a match factor greater than 90 were selected. The relative percentage content of the chemical components of AVEO were calculated using the peak-area normalization method.

### 2.4. Antioxidant Activity In Vitro of AVEO

The antioxidant capacity in vitro of AVEO was determined using radical scavenging activity (DPPH, ABTS, superoxide anion) and ferric ion reducing antioxidant power (FRAP) assays. In terms of the protocol of Bruna et al. with slight modifications [20], the DPPH radical scavenging capacity was executed. Based on the method by Hossen et al. with minor modifications [21], the ABTS radical scavenging ability of essential oil was carried out. Additionally, the FRAP assay of the essential oil was also used in accordance with a method outlined by Tongnuanchan et al. [22]. Moreover, the super oxygen anion scavenging action of AVEO was measured in light of the previously reported method with slight adjustments [23].

### 2.5. Deep-Frying Test

#### 2.5.1. Grouping and Preparation of Deep-Frying Oils

The sunflower oils were separated into four groups: sunflower oil without addition (SFO Control); sunflower oil with 0.01 g/kg TBHQ (TBHQ-0.01); sunflower oil with 1 g/kg AVEO (AVEO-1); and sunflower oil with 1.5 g/kg AVEO (AVEO-1.5). According to the size of the fryer, the amount of deep-frying oil was determined to be 4.5 kg in each fryer.

#### 2.5.2. Deep-Frying Procedure of Chinese *Maye*

The deep-frying process was accomplished continuously for 30 h, following the methods mentioned in the previous experiment by Wang et al. with a little revision [24]. During the frying procedure, the oil temperature was kept at 180 ± 5 °C and the frying time was 25 ± 5 s. The samples of 50 g of deep-frying oils and 100 g of frying Chinese *Maye* were taken at 0, 6, 12, 18, 24, and 30 h for the subsequent determination of the main physicochemical parameters of the frying oils and organoleptic evaluation of the frying *Maye*.

#### 2.5.3. Determination of Physical Parameters of Deep-Frying Oils

For the determination of the red value (RV) and yellow value (YV) of frying oil samples refer to the Chinese National Standard GB/T 22460-2008 [25]. In addition, the rapid viscosity analyzer (Starch Master 2, Stockholm, Sweden) was used to measure the viscosity of the frying oils. The detailed methods are as follows: the run temperature and speed of the analyzer were set at 25 °C and 160 rpm, respectively; then, 18 g of frying oils were weighed into the sample cell. After the measurement had started and the viscosity value had stopped increasing, the viscosity value was recorded.

#### 2.5.4. Determination of Chemical Parameters of Deep-Frying Oils

##### Measurement of Acid Value (AV), Malondialdehyde (MDA) Content, Total Polar Compounds (TPC), Conjugated Dienes (CD), and Conjugated Trienes (CT)

In accordance with the Chinese National Standard GB 5009.229-2016 [26], the AV of the frying oil samples was measured. The details are as follows: weigh an appropriate amount of frying oil samples into a 250 mL conical flask, add 80 mL of ethyl ether-isopropanol mixture and 4 drops of phenolphthalein indicator and shake well to dissolve the samples. Titrate the sample solution with the standard solution, record the volume of standard solution consumed, and combine with the volume of standard solution consumed in the blank test to calculate the acid value of the frying oil sample. The MDA content of the frying oil samples was determined according to the Chinese National Standard GB 5009.181-2016 [27]. This is achieved by weighing 5 g of the frying oil samples into a 100 mL conical flask, add 50 mL of trichloroacetic acid mixture and shake well, then place on a constant temperature shaker at 50 °C for 30 min, remove and cool to room temperature, then filter through a double layer of filter paper, discard the initial filtrate and set aside. Next, 5 mL of the filtrate and 5 mL of the standard series solution were placed in a 25 mL colorimetric tube, 5 mL of trichloroacetic acid mixture was added as a sample blank, 5 mL of thiobarbituric acid aqueous solution was added, mixed well, and then placed in a 90 °C water bath for 30 min, removed, and cooled to room temperature. The absorbance values of the sample solution and the standard series of solutions were measured at 532 nm for the quantitative analysis of MDA. In addition, on the basis of the Chinese National Standard GB/T 22500-2008 [28], the CD and CT of the frying oil samples were measured. An appropriate amount of fried sunflower oil was weighed and dissolved in isooctane at room temperature into a 25 mL volumetric flask, the test solution was poured into a quartz cuvette, and the absorbance of the specimen was measured at a wavelength of 232 nm and 268 nm to determine the content of CD and CT, respectively. Moreover, the TPC (%) of the vegetable frying oil samples at distinct sampling times was determined using an edible oil quality tester (Testo 270, Baden-Wurttemberg, Germany). The probe of the cooking oil quality tester is inserted into the fried sunflower oil, the TPC of the oil is measured and the result is expressed as a percentage, which can be read directly.

##### Changes in Fatty Acid Composition of Frying Oils

The methyl esterification of the fatty acids of the deep-frying oil samples was performed in terms of the Chinese National Standard GB 5009.168-2016 with a little modification [29]. Concisely, 0.5 g of sunflower oil samples and 6 mL NaOH methanol solution (0.5 mol/L) were poured into a round bottom flask, and then connected to the condensing reflux unit, and heating begun. After about 5 min of condensation reflux, 7 mL of BF_3_ methanol solution (1:4, *v*/*v*) and 6 mL of chromatographic n-hexane were added. When the flask was cooled, 30 mL of saturated NaCl solution were added. After standing and layering, the supernatant from the flask was aspirated into a glass test tube, and anhydrous Na_2_SO_4_ was added to thoroughly remove H_2_O. Finally, the supernatant from the glass test tube was aspirated into a vial for further analysis.

The chemical analysis for the fatty acid composition of the frying oil samples was completed using an Agilent technologies 7890B GC system equipped with a HP-88 column (100 m × 0.25 mm I.D., 0.2 µm F.T., Agilent, USA), according to the previously described method of Wang et al., with minor adjustments [30]. The initial temperature of the column was set at 170 °C, ranged between 170 °C and 220 °C at 4 °C/min, no insulation required, and then increased from 220 °C to 240 °C by 1 °C/min. The injector and detector temperatures were held at 250 °C and 280 °C, respectively. The carrier gases used were N_2_ (high purity) and H_2_, at a flow rate of 1.0 mL/min and 30.0 mL/min, separately. The spitless injection was used with a spit ratio of 1:50. Furthermore, 1.0 µL sample was injected.

### 2.6. Sensory Evaluation of Chinese Maye Samples

The organoleptic evaluation of fried food is particularly important due to the volatile components of essential oils, which can provide fried food with a distinctive flavor during the deep-frying procedure. The specific methods are as follows: the sensory estimate was carried out by some professional sensory evaluators, who were given an explanation of the indicators of the sensory evaluation before the experiment. The Chinese *Maye* were successively fried for 30 h in raw sunflower oil, sunflower oil with TBHQ, and sunflower oil with AVEO, respectively. Throughout the frying course, four kinds of fried Chinese *Maye* samples were obtained every 6 h and all of them were assessed by trained panelists. The sensory characteristics of appearance, taste, flavor, as well as overall acceptability, were appraised using a 10-point hedonic scale—displayed in Table 1 (8–10: extremely like, 5–7: moderately like, 3–4: neither like nor dislike, 0–2: extremely dislike)—of these Chinese *Maye* samples.

### 2.7. Statistical Analysis

The experimental results presented in the figures and tables were exhibited in the mean value ± standard deviation (SD), as well as exhibited in the mean value in the paper. In the statistical analysis, using IBM SPSS Statistics 20.0 (Armonk, NY, USA), one-way analysis of variance (ANOVA) was applied to compare the mean value among the treatment groups, where *p* < 0.05 was deemed to be significantly different and *p* < 0.01 was deemed to be a highly significant difference. In addition, all figures were drawn using GraphPad Prism 8.0 (San Diego, CA, USA) and all tables were produced using Microsoft Excel 2016 (Redmond, WA, USA).

## 3. Results and Discussion

### 3.1. Yield and Chemical Composition of AVEO

As the main active ingredient of *A. villosum* Lour., the essential oil has attracted widespread attention from researchers due to its unique aromatic odor and physiological properties. In the present study, the yield of the AVEO was 1.03% (*w*/*w*). At room temperature, the essential oil was yellow, opaque, and semi-solid, with a woody and cool aroma. During the GC-MS analysis, as revealed in Table 2, 41 natural volatile compounds were identified in the essential oil, accounting for 98.45% of the total components. These compounds were divided into five categories: terpenes (22 species, 22.20%), esters (6 species, 20.27%), alcohols (8 species, 3.03%), ketones (3 species, 52.79%), and aromatics (2 species, 0.16%), among which *l*-Camphor (52.65%), Bornyl acetate (19.91%), Camphene (7.42%), D-Limonene (5.99%), *β*-Myrcene (4.09%), and *α*-Pinene (1.99%) were the main chemical constituents of the AVEO. It is worth noting that camphor has a high melting point (approximately 180 °C) and is solid at room temperature, which is probably the main reason why AVEO is semi-solid at room temperature. In addition, previous research has shown that bornyl acetate has antioxidant, anti-inflammatory, anti-canner, and antiabortion activities, and is often used as a food additive and flavoring agent [31]. The antioxidant effect of AVEO may be attributed to bornyl acetate. Zhang et al. also used steam distillation to extract AVEO and then analyzed its composition [32], demonstrating that the primary compounds of the essential oil were: bornyl acetate, camphor, limonene, camphene, and *β*-caryophyllene. In addition, AVEO grown in Vietnam was extracted and its chemical constituents were identified through GC-MS and GC-FID techniques, while the monoterpene hydrocarbons represented by *β*-pinene (34.7–56.6%) and *α*-pinene (11.6–22.1%) were found to be the principal components of the essential oil [18]. Furthermore, Lu et al. similarly extracted the essential oil from the fruits of *A. villosum* Lour. from Jinping County (Yunnan Province, China) through steam distillation [16], and the results showed that its main compounds were bornyl acetate (54.54%), camphor (17.92%), camphene (6.757%), and limonene (5.249%). The differences in the main constituents and content of AVEO are probably due to the different varieties of *A. villosum* Lour. and their growing conditions. As previously reported, the extraction yields and chemical components of plant essential oils are significantly related to a number of objective factors, for instance, the plant variety, growing environment, and extraction method [30].

### 3.2. Antioxidant Activity In Vitro of AVEO

Previous studies have already shown that the leaves and roots extracts of *A. villosum* Lour. have certain antioxidant activity, but there is little research on the antioxidant properties of AVEO [13]. In this work, the antioxidant activity in vitro of AVEO was determined using DPPH, ABTS, FRAP, and superoxide anion scavenging activity assays. The DPPH radical scavenging assay is commonly applied to the in vitro antioxidant evaluation of antioxidant components in extracts, where antioxidants bind by donating hydrogen or electrons to free radicals in order to scavenge them. As depicted in Figure 1A, the DPPH radical scavenging rate of the AVEO observably increased (*p* < 0.05) with the incremental concentration of the essential oil. When the AVEO was added at a concentration of 10 mg/mL, the DPPH radical scavenging rate reached 44.82%. However, the radical scavenging rate of BHT at a concentration of 0.08 mg/mL was as high as 96.1%, which was significantly different from the AVEO (*p* < 0.05). The ABTS+ radical ion solution was oxidized to a blue-green color and the reaction system was discolored by the addition of antioxidant. By detecting the change in absorbance, the degree of antioxidant activity can be evaluated. Based on results from Figure 1B, the ABTS radical scavenging rate of the AVEO increased markedly (*p* < 0.05) as the concentration of the essential oil increased, reaching 79.97% of the scavenging rate of ABTS free radicals at a concentration of 25 mg/mL, which was not significantly different from 0.08 mg/mL BHT, which scavenged 80.62% (*p* > 0.05). The FRAP assay is a method of measuring the antioxidant capacity of a sample under low pH conditions using ferrous ions to produce a blue-purple complex (Fe^2+^-TPTZ) with TPTZ (tripyridyl triazine). As exhibited in Figure 1C, the FRAP capacity is positively correlated with the concentration of essential oil added (*p* < 0.05). Although there was still a significant difference between the ferric-reducing antioxidant power ability of the essential oil (0.69 mmol/L) at 50 mg/mL and BHT (0.74 mmol/L) at 0.08 mg/mL, the values are already very close. In the superoxide anion scavenging assay, the phthalates slowly release superoxide anions under alkaline conditions and the essential oils scavenge free radicals by binding to them or supplying electrons. According to Figure 1D, the superoxide anion radical scavenging rate of the AVEO increased from 33.38% to 63.57% with the increasing concentrations of the added essential oils, but the radical scavenging capacity of the AVEO was not as high as that of BHT, which has excellent antioxidant effects (*p* < 0.05). In conclusion, the AVEO has certain in vitro antioxidant activity and is a natural antioxidant.

### 3.3. Determination of Physical Parameters of Deep-Frying Oils

Color is an intuitive and more subjective indicator of the quality of frying oils. In general, the deepening of color in vegetable oils during deep-frying is principally a result of the formation of non-enzymatic browning compounds, which is caused by the oxidative degradation of triglycerides and free fatty acids in frying oils [33]. The changes in the RV and YV of the deep-frying oil samples are demonstrated in Figure 2A,B, respectively. The RV and YV of the four groups of sunflower oil samples augmented significantly throughout the frying process (*p* < 0.05), with the control group having the highest RV and YV compared to the other three groups. In addition, at the concentration of 1.5 g/kg of AVEO, the RV and YV of the frying oil samples were lower than the other groups, indicating that the antioxidant effect of the AVEO was better at this additional concentration, while the RV and YV of the frying oil samples with the addition of 1 g/kg AVEO were comparable to the 0.01 g/kg TBHQ. Babiker et al. separately added rosemary essential oil and extract to sunflower oil for frying, and both frying oil samples showed an upward trend in their RV and YV [34]; however, the enhancement in RV and YV was smaller in the frying oil with the essential oil than with the extract, which shows that the effect of plant essential oil on inhibiting oxidative rancidity during the frying process of sunflower is better than that of plant extract. It is widely known that viscosity is a crucial physical index to evaluate the changes in the quality of frying oils. During the high-temperature continuous frying process, the viscosity of vegetable frying oils gradually increases and is significantly higher after 6 h due to the thermal oxidation and polymerization of the deep-frying oils [35]. As shown in Figure 2C, the viscosity of the four groups of frying oil samples dramatically increased (*p* < 0.05) during the successive deep-frying process, with the control group showing the largest increment in viscosity, followed by the 1 g/kg AVEO group, the 0.01 g/kg TBHQ group, and the 1.5 g/kg AVEO group. After 12 h of deep-frying, the viscosity of both the TBHQ and essential oil samples significantly decreased, in contrast to the control group sample (*p* < 0.05). Furthermore, the addition of 1 g/kg of AVEO and 0.01 g/kg of TBHQ had a similar effect on the viscosity of the frying oils, while the addition of AVEO at a concentration of 1.5 g/kg displayed a splendid inhibitory action on the increase in the viscosity of the frying oils.

### 3.4. Determination of Chemical Parameters of Deep-Frying Oils

#### 3.4.1. Effects of AVEO on AV, MDA Content, TPC, CD, and CT of Frying Oils

As is well-known, the AV is a considerable indicator for estimating the degree of oxidative rancidity of vegetable oil. During the heating process of frying oils at high temperatures, triglycerides undergo hydrolysis reactions to form free fatty acids, which are prone to oxidation phenomena [36]. Moreover, the unsaturated esters in the frying oils undergo oxidation to produce peroxides, which are unstable and decompose into small molecules of aldehydes and ketones, and then continue to oxidize to produce acids, contributing to an enhancement of the acidity of the frying oils [37]. In Figure 3A, the AV of the four sunflower oil samples clearly increased in pace with the increase in frying time (*p* < 0.05), while the control group exhibited a higher AV than the other three groups. Additionally, 1.5 g/kg of AVEO and 0.01 g/kg of TBHQ showed a consistent inhibition of the increment in AV of the frying oil samples; 1 g/kg of AVEO presented weaker inhibition compared with these two groups. Vegetable oil undergoes an oxidative rancidity reaction under the action of light, heat, and oxygen, as a result, it decomposes into aldehydes, acids, and other compounds, while MDA is one of the decomposition products. The degree of the oxidative rancidity of vegetable oil can be deduced by measuring the content of MDA [38,39]. The content of MDA in the frying oil at different times is shown in Figure 3B. Throughout the deep-frying process, the MDA content values of the control group were generally higher than the other three groups of antioxidant-added sunflower oil samples. After 30 h of continuous frying, the MDA content values of the control, 0.01 g/kg TBHQ, 1 g/kg AVEO, and 1.5 g/kg AVEO groups were 2.5227, 2.05, 1.9176, and 1.584, respectively. It is worth noting that the higher concentration of essential oils may have promoted oxidation. During the consecutive deep-frying procedure, the oil molecules undergo oxidation, hydrolysis, cleavage, and polymerization to produce a large number of polar compounds, such as carbonyl, carboxyl, aldehyde, and ketone groups, the sum of which is known as the TPC [40]. These polar compounds are characterized by their high molecular weight and as being non-volatile; therefore, TPC is considered one of the more stable and uppermost indicators for estimating the quality of frying oils [41]. As exhibited in Figure 3C, the TPC content of the four sunflower oil frying samples significantly increased in the wake of the increase in the frying time (*p* < 0.05). After 30 h of the frying procedure, the TPC content of the control group of frying oil samples increased by 12.17%, while the frying oil samples from the group with the addition of 1.5 g/kg AVEO increased by 7.5%, which was significantly lower compared to the former (*p* < 0.05). The TPC content of the frying oil samples increased by 10.17% and 9.17% after 30 h of frying in the 0.01 g/kg TBHQ and 1.0 g/kg AVEO groups, respectively. The extent of the primary and secondary oxidation of the vegetable oils during high-temperature deep-frying is mainly indicated by the absorbance at 232 nm and 268 nm, corresponding to the products of CD and CT. CD are produced when unsaturated fatty acids in vegetable oils are oxidized to form more stable free radicals, while CT are primarily applied to characterize the changes in ketone content during the secondary oxidation of vegetable oils [42,43]. As displayed in Figure 3D, E, as the frying time continues, the content of CD and CT in the four groups of sunflower oil samples show an increasing trend, with the highest content of CD and CT in the control group, reaching 27.181 and 5.831 after 30 h of the continuous deep-frying procedure. Meanwhile, the production of CD and CT in the sunflower oils with the addition of 1.5 g/kg of AVEO reduced to 17.050 and 4.229 after 30 h of continuous frying, respectively. The effect of 1.5 g/kg of AVEO on the production of CD by the frying oils was more consistent with that of 0.01 g/kg of TBHQ, while for the production of CT, the influence of 1.5 g/kg of AVEO was stronger than that of 0.01 g/kg of TBHQ. This result revealed that the higher concentration of AVEO has a better antioxidant effect on sunflower oil and can effectively retard the oxidative rancidity of the frying oil within a certain concentration range.

#### 3.4.2. Effects of AVEO on the Fatty Acid Composition of Frying Oils

Due to the oxidative heat reaction in the high-temperature frying process, the unsaturated degree of fatty acids in frying oil decreases, resulting in more saturated fatty acids [44]. The variations in the fatty acid composition of the frying oils are represented in Table 3; four groups of frying oil samples changed to varying degrees with the frying time. The main fatty acids that varied significantly (*p* < 0.05) between the oil samples from the control group and the 0.01 g/kg TBHQ group were: stearic acid, oleic acid, linoleic acid, and α-linolenic acid. Owing to the fact that sunflower oil is rich in unsaturated fatty acids—particularly linoleic acid and α-linolenic acid, both of which are human essential fatty acids—it is of practical importance to research the transformation of the two fatty acids during deep-frying procedure [45]. As shown in Figure 3F,G, the percentage content of linoleic acid and α-linolenic acid in the four groups of frying oil samples decreased gradually throughout the deep-frying process, with the frying oil samples without antioxidant showing the most significant reduction in their content of the two fatty acids, by 4.487% and 0.062%, respectively. The effects of 1.5 g/kg AVEO and 0.01 g/kg TBHQ on the content of these two fatty acids were the more consistent, while 1 g/kg AVEO had a weaker influence. The research results indicate that 1.5 g/kg AVEO can impair the destruction of nutrients and the deterioration of the quality of the sunflower oils.

### 3.5. Sensory Evaluation of Chinese Maye

For the successive deep-frying procedure, the degree of the oxidative rancidity of frying oil is absolutely principal to the flavor of the fried product. When the organoleptic judgement scores of the fried food are terrible, it means that the quality of the vegetable oil is poor; in other words, the oil is no longer suitable for deep-frying. Thus, the sensory evaluation of frying products is quite essential [46]. The sensory evaluation scores of the fried Chinese *Maye* in terms of its appearance, taste, flavor, and overall acceptability are displayed in Table 4. Throughout the deep-frying course, the organoleptic judgement values of the Chinese *Maye* fried in sunflower oil without antioxidant gradually decreased as the frying time increased, while the degree of the reduction in the 0.01 g/kg TBHQ group is smaller than that in the control group. In addition, the sensory estimate scores for the appearance, taste, flavor, and overall acceptability of the Chinese *Maye* after 30 h of continuous deep-frying in sunflower oil with 1.5 g/kg AVEO were 5.78, 5.64, 5.84, and 5.58, respectively. In this group, the organoleptic analysis scores were higher than the control group at each sampling moment during the whole frying procedure, as well as being closer to the frying Chinese *Maye* with 0.01 g/kg TBHQ. The results show that the addition of a certain amount of AVEO can slow down the oxidative rancidity of deep-frying oils and endow the favorable sensory properties of fried Chinese *Maye*.

## 4. Conclusions

In summary, AVEO shows certain antioxidant activity in vitro, including free radical scavenging capacity and reducing ability. During the 30 h continuous deep-frying procedure, not only did the addition of AVEO at 1.5 g/kg to sunflower oil weaken the increment for the values of the AV, MDA content, TPC, CD, and CT of the frying oils, but also prevented variations in its color, viscosity, and fatty acid profile. Furthermore, the fried products obtained from the frying process of the sunflower oil flavored with AVEO exhibited preferable appearance, taste, flavor, and overall acceptability. Above all, the addition of AVEO to sunflower oil plays an extremely crucial role in increasing its excellent oxidative stability and pleasant sensory properties. The research provides a theoretical basis for the development and utilization of AVEO as a natural antioxidant and flavoring agent, as well as a direction of thought for the production of flavored oils and the cooking of flavored products.

## Figures and Tables

**Figure 1 antioxidants-12-01429-f001:**
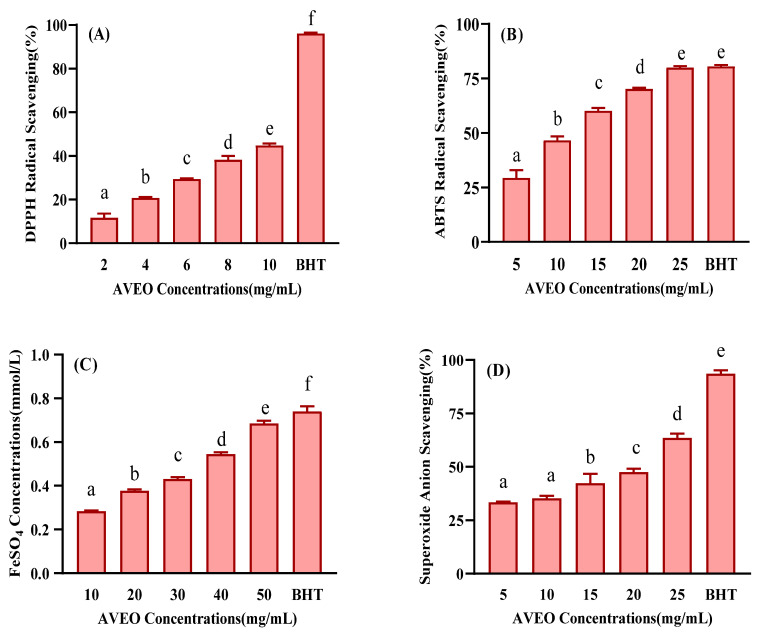
Antioxidant effects in vitro of AVEO, determined by (**A**) DPPH, (**B**) ABTS, (**C**) FRAP, and (**D**) superoxide anion scavenging effect assays, compared with BHT (positive control; 0.08 mg/mL). Values are expressed as mean ± SD (*n* = 3). In addition, “a, b, c, d, e, f” represent the variability between different groups of data, with the same letter indicating no significant difference and different letters indicating a significant difference.

**Figure 2 antioxidants-12-01429-f002:**
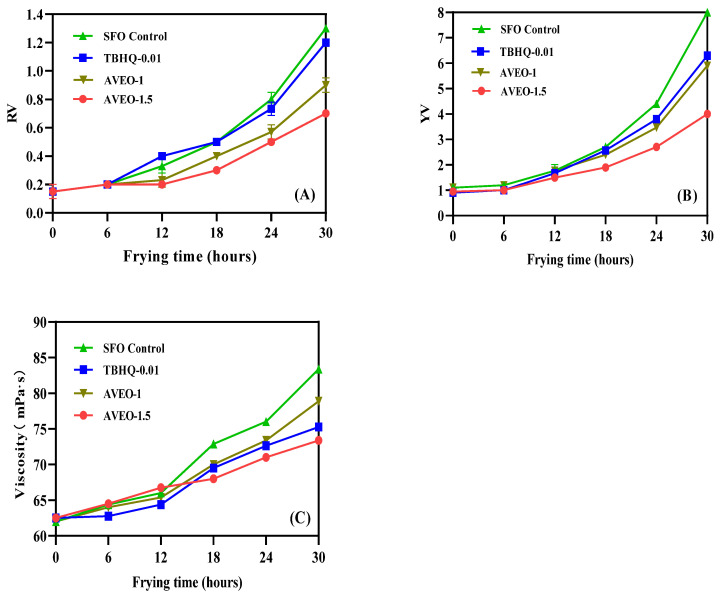
The influences of AVEO and positive TBHQ on RV (**A**), YV (**B**), and Viscosity (**C**) of sunflower oil samples during the frying process at 180 °C for 30 h. Values are expressed as mean ± SD (*n* = 3).

**Figure 3 antioxidants-12-01429-f003:**
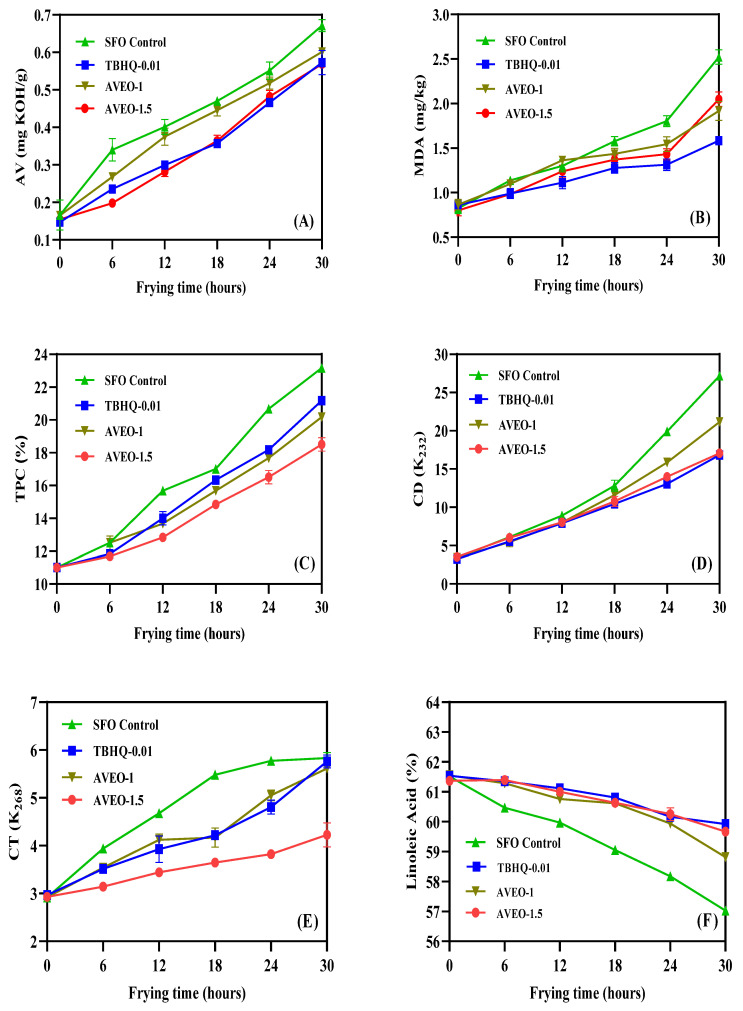
The influences of AVEO and positive TBHQ on AV (**A**), MDA (**B**), TPC (**C**), CD (**D**), CT (**E**), linoleic acid (**F**), and α-linolenic acid (**G**) of sunflower oil samples during frying process at 180 °C for 30 h. Values are expressed as mean ± SD (*n* = 3).

**Table 1 antioxidants-12-01429-t001:** The sensory scoring standards of Chinese *Maye* samples.

Items	Evaluation Criterion	Score
Appearance	golden color, clean surface, good shape	8–10
	golden color, a little oil stains on the surface, good shape	5–7
	pale yellow in color, more oil stains on the surface, fair shape	3–4
	pale yellow in color, a lot of oil stains on the surface, poor shape	0–2
Taste	crispy texture, medium firmness	8–10
	slightly crispy texture, medium firmness	5–7
	slightly crispy texture, slightly firm or soft	3–4
	poor texture, very firm or soft	0–2
Flavor	strong aroma on the nose and on the palate	8–10
	slightly fried aroma on the nose and flavor on the palate	5–7
	no flavor on the nose and on the palate	3–4
	smells like food spoilage	0–2
Overall acceptability	excellent overall acceptance	8–10
	good overall acceptance	5–7
	average overall acceptance	3–4
	poor overall acceptance	0–2

**Table 2 antioxidants-12-01429-t002:** Chemical composition of AVEO.

No.	Compound Name	Molecular Formula	CAS Number	Peak Area (%) ^a^
1	Tricyclene	C_10_H_16_	508-32-7	0.21
2	*α*-Phellandrene	C_10_H_16_	99-83-2	0.27
3	*α*-Pinene	C_10_H_16_	80-56-8	1.99
4	Camphene	C_10_H_16_	79-92-5	7.42
5	*β*-Pinene	C_10_H_16_	127-91-3	0.43
6	*β*-Myrcene	C_10_H_16_	123-35-3	4.09
7	Butyl acetate	C_6_H_12_O_2_	123-86-4	0.01
8	*α*-Terpinene	C_10_H_16_	99-86-5	0.09
9	*p*-Cymene	C_10_H_14_	99-87-6	0.08
10	*o*-Cymene	C_10_H_14_	527-84-4	0.08
11	D-Limonene	C_10_H_16_	5989-27-5	5.99
12	Eucalyptol	C_10_H_18_O	470-82-6	0.12
13	trans-*β*-Ocimene	C_10_H_16_	3779-61-1	0.10
14	*β*-Ocimene	C_10_H_16_	13877-91-3	0.02
15	*γ*-Terpinene	C_10_H_16_	99-85-4	0.09
16	(+)-4-Carene	C_10_H_16_	29050-33-7	0.06
17	Fenchone	C_10_H_16_O	1195-79-5	0.07
18	D-Fenchone	C_10_H_16_O	4695-62-9	0.07
19	Linalool	C_10_H_18_O	78-70-6	1.27
20	*l*-Camphor	C_10_H_16_O	464-48-2	52.65
21	Terpinen-4-ol	C_10_H_18_O	562-74-3	0.08
22	*α*-Terpineol	C_10_H_18_O	98-55-5	0.21
23	Acetic acid octyl ester	C_10_H_20_O_2_	112-14-1	0.05
24	Fenchyl acetate	C_12_H_20_O_2_	13851-11-1	0.03
25	Bornyl acetate	C_12_H_20_O_2_	76-49-3	19.91
26	Neryl acetate	C_12_H_20_O_2_	141-12-8	0.22
27	Acetic acid lavandulyl ester	C_12_H_20_O_2_	25905-14-0	0.05
28	Elemene	C_15_H_24_	33880-83-0	0.03
29	Caryophyllene	C_15_H_24_	87-44-5	0.35
30	Sabinene	C_10_H_16_	3387-41-5	0.02
31	(E)-*β*-Famesene	C_15_H_24_	18794-84-8	0.05
32	*cis*-*β*-Farnesene	C_15_H_24_	28973-97-9	0.07
33	*α*-Muurolene	C_15_H_24_	31983-22-9	0.07
34	Humulene	C_15_H_24_	6753-98-6	0.05
35	*β*-Bisabolene	C_15_H_24_	495-61-4	0.42
36	(+)-*δ*-Cadinene	C_15_H_24_	483-76-1	0.35
37	*β*-Sesquiphellandrene	C_15_H_24_	20307-83-9	0.03
38	*cis*-Nerolidol	C_15_H_26_O	142-50-7	0.15
39	Nerolidol	C_15_H_26_O	40716-66-3	0.15
40	*α*-Cadinol	C_15_H_26_O	481-34-5	0.38
41	*β*-Bisabolol	C_15_H_26_O	15352-77-9	0.67
Total				98.45

^a^ The relative percentage content of essential oil constituents is expressed as peak area.

**Table 3 antioxidants-12-01429-t003:** Variation of the fatty acid composition of frying oil at 180 °C for 30 h influenced by AVEO ^a^.

Group	Frying Time (h)	C14:0 Myristic Acid	C16:0 Palmitic Acid	C16:1 ω7 Palmitoleic Acid	C18:0 Stearic Acid	C18:1 ω9 Oleic Acid	C18:2 ω6 Linoleic Acid	C20:0 Arachidic Acid	C18:3 ω3 α-Linolenic Acid	C22:0 Behenic Acid
AVEO (1.5 g/kg)	0	0.077 ± 0.001	6.292 ± 0.059	0.079 ± 0.007	3.744 ± 0.001	26.989 ± 0.004	61.573 ± 0.051	0.242 ± 0.002	0.259 ± 0.001	0.746 ± 0.002
	6	0.075 ± 0.001	6.323 ± 0.007	0.079 ± 0.001	3.771 ± 0.001	27.123 ± 0.008	61.396 ± 0.004	0.240 ± 0.002	0.244 ± 0.005	0.751 ± 0.002
	12	0.077 ± 0.001	6.496 ± 0.002	0.075 ± 0.001	3.827 ± 0.013 ^b^	27.285 ± 0.026 ^b^	61.002 ± 0.006 ^b^	0.241 ± 0.005	0.234 ± 0.001 ^b^	0.765 ± 0.001
	18	0.077 ± 0.004	6.571 ± 0.005	0.075 ± 0.001	3.844 ± 0.001 ^b^	27.573 ± 0.039 ^b^	60.635 ± 0.046 ^b^	0.241 ± 0.002	0.217 ± 0.004 ^b^	0.770 ± 0.011
	24	0.078 ± 0.001	6.739 ± 0.289	0.075 ± 0.001	3.855 ± 0.006 ^b^	27.766 ± 0.091 ^b^	60.255 ± 0.214 ^b^	0.243 ± 0.001	0.208 ± 0.004 ^b^	0.779 ± 0.014
	30	0.079 ± 0.001 ^b^	6.893 ± 0.134 ^b^	0.076 ± 0.002	3.933 ± 0.021 ^b^	28.108 ± 0.027 ^b^	59.668 ± 0.101 ^b^	0.248 ± 0.004	0.203 ± 0.001 ^b^	0.794 ± 0.014 ^b^
AVEO (1 g/kg)	0	0.069 ± 0.001	6.476 ± 0.190	0.074 ± 0.003	3.781 ± 0.011	26.767 ± 0.007	61.549 ± 0.161	0.256 ± 0.011	0.249 ± 0.001	0.784 ± 0.003 ^c^
	6	0.070 ± 0.001	6.472 ± 0.006	0.075± 0.007	3.832 ± 0.004 ^bc^	26.983 ± 0.011 ^b^	61.290 ± 0.021	0.248 ± 0.001	0.244 ± 0.006	0.781 ± 0.009 ^c^
	12	0.074 ± 0.001	6.447 ± 0.002	0.077 ± 0.001	3.888 ± 0.005 ^b^	27.440 ± 0.068 ^b^	60.766 ± 0.050 ^bc^	0.251 ± 0.001	0.245 ± 0.001	0.815 ± 0.011 ^c^
	18	0.074 ± 0.003	6.527 ± 0.007	0.071 ± 0.008	3.928 ± 0.004 ^bc^	27.462 ± 0.028 ^b^	60.619 ± 0.009 ^b^	0.272 ± 0.001 ^c^	0.239 ± 0.001 ^bc^	0.811 ± 0.016
	24	0.075 ± 0.001	6.659 ± 0.001	0.077 ± 0.001	4.014 ± 0.004 ^bc^	27.921 ± 0.008 ^bc^	59.946 ± 0.021 ^b^	0.257 ± 0.001 ^c^	0.216 ± 0.001 ^bc^	0.837 ± 0.034 ^c^
	30	0.077 ± 0.001 ^b^	6.814 ± 0.013 ^b^	0.082 ± 0.001	4.054 ± 0.010 ^bc^	28.838 ± 0.097 ^bc^	58.819 ± 0.106 ^bc^	0.263 ± 0.001 ^c^	0.207 ± 0.001 ^b^	0.849 ± 0.033
TBHQ (0.01 g/kg)	0	0.076 ± 0.001	6.405 ± 0.099	0.077 ± 0.004	3.723 ± 0.004	26.931 ± 0.068	61.534 ± 0.025	0.251 ± 0.002	0.254 ± 0.002	0.751 ± 0.003
	6	0.076 ± 0.002	6.380 ± 0.270	0.075 ± 0.002	3.740 ± 0.029	27.189 ± 0.168	61.355 ± 0.060 ^b^	0.244 ± 0.001	0.251 ± 0.005	0.754 ± 0.005
	12	0.077 ± 0.001	6.386 ± 0.002	0.072 ± 0.004	3.792 ± 0.014 ^b^	27.208 ± 0.044	61.118 ± 0.041 ^b^	0.242 ± 0.002	0.239 ± 0.006	0.762 ± 0.002
	18	0.077 ± 0.001	6.524 ± 0.001	0.070 ± 0.003	3.825 ± 0.011 ^b^	27.446 ± 0.014 ^b^	60.809 ± 0.018 ^bc^	0.247 ± 0.001 c	0.231 ± 0.009 ^bc^	0.773 ± 0.008 ^b^
	24	0.078 ± 0.001	6.693 ± 0.006	0.077 ± 0.001	3.980 ± 0.002 ^bc^	27.780 ± 0.003 ^b^	60.168 ± 0.011 ^b^	0.245 ± 0.001	0.215 ± 0.002 ^bc^	0.763 ± 0.001 ^b^
	30	0.078 ± 0.001	6.722 ± 0.001	0.080 ± 0.014	3.938 ± 0.006 ^b^	28.034 ± 0.049 ^b^	59.924 ± 0.030 ^b^	0.249 ± 0.003	0.204 ± 0.002 ^b^	0.774 ± 0.001 ^b^
SFO Control	0	0.075 ± 0.003	6.505 ± 0.004	0.074 ± 0.001	3.815 ± 0.040	26.740 ± 0.009	61.517 ± 0.032	0.249 ± 0.002	0.254 ± 0.002	0.772 ± 0.001 ^c^
	6	0.079 ± 0.001	6.697 ± 0.035 ^b^	0.074 ± 0.001	3.839 ± 0.005 ^bc^	27.559 ± 0.006 ^b^	60.473 ± 0.035 ^bc^	0.255 ± 0.001 ^bc^	0.250 ± 0.002 ^b^	0.777 ± 0.002 ^bc^
	12	0.078 ± 0.002	6.824 ± 0.011 ^bc^	0.078 ± 0.002	3.973 ± 0.020 ^b^	27.816 ± 0.033 ^bc^	59.968 ± 0.034 ^bc^	0.260 ± 0.001 ^bc^	0.217 ± 0.002 ^bc^	0.786 ± 0.002 ^bc^
	18	0.081 ± 0.001	6.930 ± 0.076 ^bc^	0.082 ± 0.002 ^b^	4.086 ± 0.025 ^bc^	28.481 ± 0.012 ^bc^	59.055 ± 0.037 ^bc^	0.267 ± 0.002 ^bc^	0.210 ± 0.001 ^b^	0.808 ± 0.002 ^b^
	24	0.084 ± 0.002 ^c^	7.233 ± 0.001 ^bc^	0.082 ± 0.002 ^bc^	4.196 ± 0.00 ^bc^	29.036 ± 0.026 ^bc^	58.174 ± 0.028 ^bc^	0.278 ± 0.001 ^bc^	0.202 ± 0.002 ^b^	0.827 ± 0.001 ^b^
	30	0.087 ± 0.001 ^bc^	7.547 ± 0.002 ^bc^	0.087 ± 0.001 ^b^	4.276 ± 0.016 ^bc^	29.710 ± 0.002 ^bc^	57.030 ± 0.021 ^bc^	0.284 ± 0.002 ^bc^	0.192 ± 0.002 ^bc^	0.835 ± 0.001 ^b^

^a^ Values are expressed as mean ± SD (*n* = 3). ^b^ As compared to the same group on the 0 h: *p* < 0.05. ^c^ As compared to the SFO control group at the same time: *p* < 0.05.

**Table 4 antioxidants-12-01429-t004:** Sensory evaluation score of Chinese *Maye*
^a^.

Items	Frying Time (h)	SFO Control	TBHQ-0.01	AVEO-1.5
Appearance	0	8.71 ± 0.82	8.71 ± 0.82	8.71 ± 0.82
	6	8.03 ± 0.74	8.27 ± 0.91	8.54 ± 0.84
	12	7.22 ± 0.68	7.43 ± 1.48	7.60 ± 0.69
	18	6.37 ± 0.84	6.12 ± 0.86	7.69 ± 0.60
	24	5.65 ± 0.75	5.80 ± 0.82	6.45 ± 0.75 ^b^
	30	4.78 ± 0.77	5.54 ± 0.57 ^b^	5.78 ± 0.72 ^b^
Taste	0	8.85 ± 1.40	8.85 ± 1.40	8.85± 1.40
	6	8.13 ± 0.93	8.43 ± 0.91	8.20 ± 0.65
	12	7.25 ± 1.54	7.78 ± 0.84	7.89 ± 0.94
	18	6.47 ± 1.34	6.50 ± 0.76	7.24 ± 1.12
	24	5.49 ± 0.95	5.64 ± 0.53	6.06 ± 0.98
	30	4.84 ± 0.78	5.05 ± 1.11	5.64 ± 0.89 ^b^
Flavor	0	8.64 ± 1.21	8.64 ± 1.21	8.64 ± 1.21
	6	8.15 ± 0.84	8.45 ± 0.65	8.35 ± 0.80
	12	7.22 ± 0.95	7.50 ± 0.83	7.85 ± 0.55
	18	6.26 ± 1.53	6.32 ± 0.66	7.01 ± 0.79
	24	5.43 ± 0.67	5.65 ± 1.21	6.42 ± 0.66 ^b^
	30	4.21 ± 0.83	4.97 ± 0.65	5.84 ± 1.04 ^b^
Overall acceptability	0	8.91 ± 1.04	8.91 ± 1.04	8.91 ± 1.04
	6	7.75 ± 0.96	7.85 ± 0.76	8.26 ± 0.70
	12	6.87 ± 0.89	6.46 ± 1.41	7.77 ± 0.90
	18	5.28 ± 0.90	5.78 ± 0.66	7.26 ± 1.26 ^b^
	24	4.09 ± 1.22	4.63 ± 0.70	6.61 ± 0.77 ^c^
	30	3.30 ± 0.83	4.11 ± 0.46	5.58 ± 0.83 ^c^

^a^ Values are expressed as mean ± SD (*n* = 6). ^b^ As compared to the SFO control group at the same time: *p* < 0.05. ^c^ As compared to the SFO control group at the same time: *p* < 0.01.

## Data Availability

The data are available upon request from the corresponding author.

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
