# Peer review of "Enhancement of Oxidative Stability of Deep-Fried Sunflower Oil by Addition of Essential Oil of Amomum villosum Lour."

_antioxidants, 2023, doi:10.3390/antiox12071429_

Round 1

Reviewer 1 Report

Response to Antioxidant 30062023

The manuscript presented some interesting ideas about cooking oil , antioxidants and others. The data presented is interesting and important for food. There are a couple of inconsistencies in the current manuscript where the authors need to address. 

1)     After reading the introduction, results and finally conclusion, it is a little hard to relate how the antioxidant of AVEO comes into play. Nevertheless, the results are truly of interest to many parties as it tries to relate what may happen through frying and cooking of food. 

2)    For the GC-MS analysis of AVEO , it was unclear how well the method was validated. It seems that normalized peak area/height were used. External calibration was not performed. It was unclear how values in Table 1 were obtained from w/w %. Values in w/w% can only be obtained from normalised vales when response factors were determined.

3)    Is there any batch to batch variability for the AVEO extracts from steam distillation. 

4)    Antioxidant properties of the AVEO extracts were really poor, it seems that there is little or even no antioxidant properties (Figure 1). It seems that extracts will likely to have little or no protective effects. 

5)    For all figures and tables, the number samples used (n=3 or 6) need to be included.

NA

Author Response

Dear Editor and Reviewers,

Thank you for your letter and for the reviewers’ comments concerning our manuscript entitled “Characterization of Antioxidant Activity and Capacity to Improve the Oxidative Stability of Sunflower Oil during Continuous Deep-Frying of the Essential Oil of Amomum villosum Lour.” (ID: antioxidants-2495673). These comments are all valuable and very helpful for revising and improving our article, as well as the important guiding significance to our researches. We have studied comments carefully and have made point-by-point responses to the comments from each reviewer. Revised portions in the manuscript are highlighted by using red color in MS Word. The responses to the reviewer’s comments are as follows:

Response to Reviewer 1 Comments:

The manuscript presented some interesting ideas about cooking oil, antioxidants and others. The data presented is interesting and important for food. There are a couple of inconsistencies in the current manuscript where the authors need to address.

Response: Thanks for the comment.

Point 1: After reading the introduction, results and finally conclusion, it is a little hard to relate how the antioxidant of AVEO comes into play. Nevertheless, the results are truly of interest to many parties as it tries to relate what may happen through frying and cooking of food.

Response: Yes, Thank you for your comments. “Previous research has shown that bornyl acetate has antioxidant, anti-inflammatory, anti-canner and antiabortion activities and is often used as a food additive and flavoring agent [26]. The antioxidant effect of AVEO may be attributed to bornyl acetate.”

  1. Wang, H.; Ma, D.; Yang, J.; Deng, K.; Li, M.; et al. An integrative volatile terpenoid profiling and transcriptomics analysis for gene mining and functional characterization of AvBPPS and AvPS involved in the monoterpenoid biosynthesis in Amomum villosum. Front. Plant Sci. 2018, 9, 846.

This section has been added to the article.

Point 2: For the GC-MS analysis of AVEO, it was unclear how well the method was validated. It seems that normalized peak area/height were used. External calibration was not performed. It was unclear how values in Table 1 were obtained from w/w%. Values in w/w% can only be obtained from normalized vales when response factors were determined.

Response: Yes. Thanks for putting this question. In the GC-MS analysis of AVEO, the assay was validated well after repeated experiments revealed no significant differences in the type and content of the chemical composition of the essential oil. The relative percentages (w/w%) of the chemical components identified in essential oils are calculated as a percentage of the total peak area in the chromatogram (peak area normalization) and are not determined by internal or external standardization.

Point 3: Is there any batch to batch variability for the AVEO extracts from steam distillation?

Response: Yes, Thank you for your comments. During the steam distillation process under the same conditions, there was minor variability in the extracts of different batches of the same species of Sha Ren (including essential oil extraction rate, essential oil composition, antioxidant effect), but these differences were insignificant and had little effect on the experimental results.

Point 4: Antioxidant properties of the AVEO extracts were really poor, it seems that there is little or even no antioxidant properties (Figure 1). It seems that extracts will likely to have little or no protective effects.

Response: Yes, Thanks for your comments. During the in vitro antioxidant assay (Figure 1), we indeed found that the antioxidant effect of the AVEO was not as good as that of the synthetic antioxidant (BHT) at the same concentration, which is a common disadvantage of natural antioxidants, but when the concentration of the essential oil was increased to a certain level, it had a similar antioxidant effect to that of a certain concentration of the synthetic antioxidant. In addition, the AVEO also showed some antioxidant effects in subsequent frying experiments, so it may be used as a natural antioxidant in the frying of food products.

Point 5: For all figures and tables, the number samples used (n=3 or 6) need to be included.

Response: Yes. Thanks for pointing out the mistake. We have added or modified the number of samples used in all figures and tables according to your comments.

Reviewer 2 Report

The manuscript deals with the characterization of antioxidant activity and capacity to improve the oxidative stability of sunflower oil during continuous deep-frying of the essential oil of Amomum villosum Lour.

The text is well-written.

Please, see and consider the following comments and suggestions for amendments. All answers should be included in the manuscript.

1.                  Ls.107-8: Please, add the exact temperature.

2.                  Ls.108-9: The temperature range in which the distillation was carried out should be added.

3.                  L.261: pH.

4.                  L.261: The “blue-purple complex” should be further defined, i.e., molecular formula, etc..

5.                  L.262: What is TPTZ?

6.                  L.344: significantly

7.                  What is the cost of AVEO added in sunflower? Compared with other antioxidants? The authors should comment on the economic aspect of their study.

Author Response

Dear Editor and Reviewers,

Thank you for your letter and for the reviewers’ comments concerning our manuscript entitled “Characterization of Antioxidant Activity and Capacity to Improve the Oxidative Stability of Sunflower Oil during Continuous Deep-Frying of the Essential Oil of Amomum villosum Lour.” (ID: antioxidants-2495673). These comments are all valuable and very helpful for revising and improving our article, as well as the important guiding significance to our researches. We have studied comments carefully and have made point-by-point responses to the comments from each reviewer. Revised portions in the manuscript are highlighted by using red color in MS Word. The responses to the reviewer’s comments are as follows:

Response to Reviewer 2 Comments:

The manuscript deals with the characterization of antioxidant activity and capacity to improve the oxidative stability of sunflower oil during continuous deep-frying of the essential oil of Amomum villosum Lour. The text is well-written. Please, see and consider the following comments and suggestions for amendments. All answers should be included in the manuscript.

Response: Thanks for the comment.

Point 1: Lines.107-108: Please, add the exact temperature. Lines.108-109: The temperature range in which the distillation was carried out should be added.

Response: Yes. Thanks for pointing out these mistakes. We have added the exact temperature (180±5 ℃). Because the same heating device does not have exactly the same heating effect, appropriate adjustments are made within this heating temperature range to keep the mixture in the round bottom flask at a slight boil in order to obtain essential oil.

Point 2: Line.261: pH. Line.261: The “blue-purple complex” should be further defined, i.e., molecular formula, etc. Line.262: What is TPTZ?

Response: Yes. Thanks for pointing out these mistakes. “PH” has been amended to “pH”. The “blue-purple complex” is Fe2+-TPTZ and the TPTZ is tripyridyl triazine. We have corrected these mistakes in the manuscript.

Point 3: Line.344: significantly

Response: Yes. Thanks for pointing out this mistake. “signally” has been amended to “significantly”.

Point 4: What is the cost of AVEO added in sunflower? Compared with other antioxidants? The authors should comment on the economic aspect of their study.

Response: Yes. Thanks for raising this issue. There has been much discussion about the cost of replacing synthetic antioxidants in food with essential oils from plants. The cost of AVEO in this experiment is indeed higher than that of synthetic antioxidants, but the food industry is now moving towards green safety, and the potential harmful effects of synthetic antioxidants on humans have led researchers to favor natural plant extracts, whose special smell and numerous biological activities would be an advantage for natural antioxidants in the food industry. The issue of cost compared to other natural antioxidants needs further research.

Reviewer 3 Report

The manuscript includes an interesting study. I think wide performances of the experimental part ought to be provided before a subsequent revision and acceptance.

The title could be simplified. It is too long as it is. Something like “Enhancement of oxidative stability of deep-fried sunflower oil by addition of essential oil of Amomum villosum lour”.

Abstract

Some more information about the study carried out ought to be included. In line 20, indicate the in vitro analyses carried out. In line 23, indicate the indices checked. In order to spare words, some shortening of lines 15-18 could be done as it is too general information.

Keywords

Include deep-frying.

Material and methods

Line 168: Malondialdehyde is not measured but its content. This kind of mistake ought to be performed throughout the whole manuscript.

Lines 170-175: Provide minor details on such technique.

Lines 180-181: Provide minor details on such technique.

Explain the qualitative and quantitative analyses.

Line 202Express clearly what the limit of acceptability is.

Lines 204-210: What is the number of replicates ?

Results

Table 1: A very interesting study has been carried out.

Figure 2: X axis: Time of what ? Please, clarify. The same in other Figures.

Figure 3: Y axis: per g or per kg of what ?

Minor performances could be done.

Author Response

Dear Editor and Reviewers,

Thank you for your letter and for the reviewers’ comments concerning our manuscript entitled “Characterization of Antioxidant Activity and Capacity to Improve the Oxidative Stability of Sunflower Oil during Continuous Deep-Frying of the Essential Oil of Amomum villosum Lour.” (ID: antioxidants-2495673). These comments are all valuable and very helpful for revising and improving our article, as well as the important guiding significance to our researches. We have studied comments carefully and have made point-by-point responses to the comments from each reviewer. Revised portions in the manuscript are highlighted by using red color in MS Word. The responses to the reviewer’s comments are as follows:

Response to Reviewer 3 Comments:

The manuscript includes an interesting study. I think wide performances of the experimental part ought to be provided before a subsequent revision and acceptance.

Response: Thanks for the comment.

Point 1: The title could be simplified. It is too long as it is. Something like “Enhancement of oxidative stability of deep-fried sunflower oil by addition of essential oil of Amomum villosum Lour.”.

Response: Yes. Thanks for raising this issue. The title of article has been amended to "Enhancement of Oxidative Stability of Deep-fried Sunflower Oil by Addition of Essential Oil of Amomum villosum Lour.”. This is truly a concise and condensed title.

Point 2: Abstract

Some more information about the study carried out ought to be included. In line 20, indicate the in vitro analyses carried out. In line 23, indicate the indices checked. In order to spare words, some shortening of lines 15-18 could be done as it is too general information.

Response: Yes. Thanks for your comments. In response to your comments, the part of abstract has been revised to add more information relevant to the article and to streamline or delete irrelevant sentences, and the relevant modifications have been highlighted in red color in the manuscript.

Point 2: Keywords

Include deep-frying.

Response: Yes. Thanks for raising this issue. “Deep-frying” has been added to Keywords.

Point 2: Materials and methods

Line 168: Malondialdehyde is not measured but its content. This kind of mistake ought to be performed throughout the whole manuscript.

Lines 170-175: Provide minor details on such technique.

Lines 180-181: Provide minor details on such technique.

Explain the qualitative and quantitative analyses.

Line 202: Express clearly what the limit of acceptability is.

Lines 204-210: What is the number of replicates?

Response: Yes. Thank you for pointing out these problems.

For line 168, “malondialdehyde (MDA)” has been corrected to “malondialdehyde (MDA) content”, and the same mistakes in the article have also been revised.

For lines 170-175: The detailed methodology for the determination of this section has been added to the article, with the following modifications: In accordance with the China National Standard GB 5009.229-2016, the AV of the frying oil samples were measured. The details are as follows: weigh an appropriate amount of frying oil samples into a 250 mL conical flask, add 80 mL of ethyl ether-isopropanol mixture and 4 drops of phenolphthalein indicator and shake well to dissolve the samples. Titrate the sample solution with the standard solution, record the volume of standard solution consumed and combine with the volume of standard solution consumed in the blank test to calculate the acid value of the frying oil sample. The MDA content of the frying oil samples were determined according to the China National Standard GB 5009.181-2016. This is done by weighing 5 g of the frying oil samples into a 100 mL conical flask, add 50 mL of trichloroacetic acid mixture and shake well, then place on a constant temperature shaker at 50 °C for 30 min, remove and cool to room temperature, then filter through a double layer of filter paper, discard the initial filtrate and set aside. 5 mL of the filtrate and 5 mL of the standard series solution were placed in a 25 mL colorimetric tube, 5 mL of trichloroacetic acid mixture was added as a sample blank, 5 mL of thiobarbituric acid aqueous solution was added, mixed well and then placed in a 90 °C water bath for 30 min, removed and cooled to room temperature. The absorbance values of the sample solution and the standard series of solutions were measured at 532 nm for the quantitative analysis of MDA. In addition, on the basis of the China National Standard GB/T 22500-2008, the CD and CT of the frying oil samples were measured. An appropriate amount of fried sunflower oil was weighed and dissolved in isooctane at room temperature into a 25 mL volumetric flask, the test solution was poured into a quartz cuvette and the absorbance of the specimen was measured at a wavelength of 232 nm and 268 nm to determine the content of CD and CT respectively. What is more, the TPC (%) of vegetable frying oil samples at distinct sampling times was determined using an edible oil quality tester (Testo 270, Germany). The probe of the cooking oil quality tester is inserted into the fried sunflower oil, the TPC of the oil is measured and the result is expressed as a percent-age, which can be read directly.

For lines 180-181: Fatty acid methyl esterification of frying oils is carried out as follows: 0.5 g of sunflower oil samples and 6 mL NaOH methanol solution (0.5 mol/L) were poured into a round bottom flask, and then connected the condensing reflux unit and started heating. After about 5 minutes of condensation reflux, 7 mL of BF3 methanol solution (1:4, v/v) and 6.0 mL of chromatographic n-hexane were added. When the flask was cooled, 30 mL of saturated NaCl solution were added. After standing and layering, the supernatant from the flask was aspirated into a glass test tube, and anhydrous Na2SO4 was added to thoroughly remove H2O. Finally, the supernatant from the glass test tube aspirated into a vial for the further analysis.

For the qualitative and quantitative analyses, the fatty acid composition of the frying oil was characterized by comparing the peak time of the gas chromatogram with the peak time of the standard curve, while the fatty acid composition was quantified using the peak area normalization method. This explanation has been added to the manuscript.

For lines 202, the specific scoring criteria and limits for the sensory evaluation are shown in Table 1 and the table has been added to the article.

Table 1. The sensory scoring standards of Chinese Maye

Items

Evaluation criterion

Score

Appearance

Golden color, clean surface, good shape

8-10

Golden color, a little oil stains on the surface, good shape

5-7

Pale yellow in color, more oil stains on the surface, fair shape

3-4

Pale yellow in color, a lot of oil stains on the surface, poor shape

0-2

Taste

Crispy texture, medium firmness

8-10

Slightly Crispy texture, medium firmness

5-7

Slightly Crispy texture, slightly firm or soft

3-4

Poor texture, very firm or soft

0-2

Flavor

Strong aroma on the nose and on the palate

8-10

Slightly fried aroma on the nose and flavor on the palate

5-7

No flavor on the nose and on the palate

3-4

Smells like food spoilage

0-2

Overall

acceptability

Excellent overall acceptance

8-10

Good overall acceptance

5-7

Average overall acceptance

3-4

Poor overall acceptance

0-2

For lines 204-210, with the exception of the sensory evaluation of fried Maye, which was repeated 6 times, all other experiments (including the analysis of the volatile components of the essential oil, the determination of the in vitro antioxidant capacity of the essential oil and the determination of the physicochemical parameters of fried sunflower oil) were repeated 3 times, and the number of repetitions of the experiments are indicated in the corresponding tables or figures.

Point 5: Results

Table 1: A very interesting study has been carried out.

Figure 2: X axis: Time of what? Please, clarify. The same in other Figures.

Figure 3: Y axis: per g or per kg of what?

Response: Yes. Thanks for putting these questions. For Figure 2, the time (X axis) represents the continuous frying time in sunflower oil. The labels of the x-axis in the paper's figures have been changed to “Frying time”. For Figure 3, the unit of acid value is mg KOH/g (representing the number of milligrams of standard titration solution KOH consumed per gram of sunflower frying oil titrated). And the MDA content is given in units of mg/kg (representing the number of milligrams of MDA per kilogram of sunflower frying oil).

Round 2

Reviewer 1 Report

1) It seems the authors may not have understood what is mean by expressing the result in w/w or normalized peak area. By expressing the result in w/w, you will have to set up external standard calibration, determine the content and calculate to the dry weight of samples used. Based on what is written , this step was not done, hence there is a serious flaw in table 2. 

2) An assay is not well validated by  well after repeated experiments revealed no significant differences in the type and content of the chemical composition of the essential oil, there are a set of criteria that needs to be followed . it seems that the authors had refer to ref 11, the work is done by another lab, the authors are required to produce relevant data to show  that the method is well validated and able to generate accurate and precise results. 

NA 

Reviewer 3 Report

The manuscript has been performed according to previous comments. I would recommend its publication.

Minor performances could be done.
